# Intraperitoneal Injection of the *Porphyromonas gingivalis* Outer Membrane Vesicle (OMV) Stimulated Expressions of Neuroinflammatory Markers and Histopathological Changes in the Brains of Adult Zebrafish

**DOI:** 10.3390/ijms252011025

**Published:** 2024-10-14

**Authors:** Malik Adewoyin, Ahmed Hamarsha, Rasaq Akinsola, Seong Lin Teoh, Mohammad Noor Amal Azmai, Noraini Abu Bakar, Nurrul Shaqinah Nasruddin

**Affiliations:** 1Department of Craniofacial Diagnostics and Biosciences, Faculty of Dentistry, Universiti Kebangsaan Malaysia, Jalan Raja Muda Abdul Aziz, Kuala Lumpur 50300, Malaysia; p111607@siswa.ukm.edu.my (M.A.); p119221@siswa.ukm.edu.my (A.H.); 2Department of Medicine, Division of Hematology and Oncology, Cedars-Sinai Medical Center, Los Angeles, CA 90048, USA; rasaq.akinsola@cshs.org; 3Department of Anatomy, Faculty of Medicine, Universiti Kebangsaan Malaysia, Jalan Yaacob Latif, Bandar Tun Razak, Cheras, Kuala Lumpur 56000, Malaysia; teohseonglin@ukm.edu.my; 4Aquatic Animal Health and Therapeutics Laboratory, Institute of Bioscience, Universiti Putra Malaysia, UPM, Serdang 43400, Selangor, Malaysia; mnamal@upm.edu.my; 5Department of Biology, Faculty of Science, Universiti Putra Malaysia, UPM, Serdang 43400, Selangor, Malaysia; norainibakar@upm.edu.my

**Keywords:** *P. gingivalis*, outer membrane vesicles, neuroinflammation

## Abstract

*Porphyromonas gingivalis* is the major pathogenic bacteria found in the subgingival plaque of patients with periodontitis, which leads to neuroinflammation. The bacteria destroy periodontal tissue through virulence factors, which are retained in the bacteria’s outer membrane vesicles (OMV). This study aimed to determine the real-time effect of an intraperitoneal injection of *P. gingivalis* OMV on the production and expression of inflammatory markers and histopathological changes in adult zebrafishes’ central nervous systems (CNS). Following the LD50 (21 µg of OMV), the zebrafish were injected intraperitoneally with 18 µg of OMVs, and the control group were injected with normal saline at seven different time points. Brains of experimental zebrafish were dissected at desired time points for colorimetric assays, ELISA, and histology. This study discovered that nitric oxide and PGE2 were significantly increased at 45 min, while IL-1β and IL-6 were expressed at subsequent 12 h and 24 h time points, respectively. Histopathological changes such as blood coagulation, astrocytosis, edema, spongiosis, and necrosis were observed between the 6hour and 24 h time points. The two apoptotic enzymes, caspases 3 and 9, were not expressed at any point. In summary, the OMV-induced neuroinflammatory responses and histopathological changes in adult zebrafish were time-point dependent. This study will enrich our understanding of the mechanism of *P. gingivalis* OMVs in neuroinflammation in a zebrafish model, most especially the timing of the expression of inflammatory mediators in relation to observable changes in brain tissues.

## 1. Introduction

Neuroinflammation is triggered by specific immune cells that inhabit the brain and are recognized for their crucial function in maintaining homeostasis in the central nervous system (CNS) as well as contributing to the pathogenesis of neurodegenerative diseases, constituting a typical hallmark of these conditions [1]. However, it has been postulated that periodontitis is correlated with neurodegenerative diseases through two distinct pathways: the first and predominant pathway pertains to the presence of inflammatory mediators in the blood circulation, thereby fostering a persistent state of inflammation; while an alternative pathway involves the role of Gram-negative bacteria, such as *Porphyromonas gingivalis* (*P. gingivalis*), which may directly precipitate tissue damage [2].

Notably, individuals with periodontal disease have higher proportions of periodontal bacteria (*Aggregatibacter actinomycetemcomitans*, *P. gingivalis*, *Tannerella forsythia*, *Treponema denticola*, *Prevotella intermedia*, and *Fusobacterium nucleatum*), higher C-reactive protein levels, and elevated proinflammatory cytokines and lymphocytes. This condition is linked to persistent neuroinflammation, microglial priming, systemic inflammation, and increased proinflammatory cytokine levels in the blood [2,3,4]. Proinflammatory cytokines produced by epithelial cells within the pathological periodontal pockets, instigated by toxic products of dysbiotic oral bacteria, can penetrate the brain parenchyma through the circulatory system. Upon entering the CNS, these cytokines stimulate the activation of resident immune cells (microglia and astrocytes), prompting their transformation into proinflammatory phenotypes, while also inducing the release of their proinflammatory mediators. Such mediators subsequently trigger signal transduction pathways that culminate in neuronal apoptosis. Should this pathological process persist over an extended duration, the resultant neuronal cell death becomes manifest as neurodegeneration [5,6,7].

The keystone periodontal bacteria, *P. gingivalis*, possesses numerous virulence factors that have been implicated in contributing to its pathogenic impact at both local and systemic levels [8]. Like most Gram-negative bacteria, *P. gingivalis* generates outer membrane vesicles (OMVs) that preserve various virulence factors from the original cell, such as LPS, fimbriae, and gingipains [9]. OMVs can worsen disease conditions by prompting the upregulation of inflammatory molecules [10]. For instance, *Helicobacter pylori* (*H. pylori*) OMVs stimulate in vitro interleukin 6 (IL-6) production in human peripheral blood mononuclear cells, while *Salmonella* spp. OMVs induce TNF and nitric oxide (NO) production in mouse macrophages [11]. Recently, it has also been found that *P. gingivalis* OMVs can trigger the production of NO and expression of inducible nitric oxide synthase (iNOS) in mouse macrophages [12].

Lately, there’s a surge in the use of zebrafish models to mimic human neurodegenerative disorders [13,14,15] and some physiological conditions that are strongly associated with neurodegeneration, such as neuroinflammation, anxiety [13,14,15,16], neuronal redox imbalance [16,17,18], and dementia [14,19,20,21,22]. Based on the link between periodontitis and neurodegenerative/systemic diseases, *P. gingivalis* and Gram-negative bacteria derivatives such as LPS [16,23,24], gingipain [25,26], and OMVs [27,28] have been used to establish relevant zebrafish disease models. We propose that a single dose of an OMV in adult zebrafish will trigger the activation of several inflammatory mediators at varying time points with corresponding impacts on the brain tissue.

This study investigated the effect of the *P. gingivalis* OMV on the upregulation of neuroinflammatory markers in real time and the relationship between the markers and histopathological alterations in the brain tissue of OMV-treated zebrafish. The outcome of this study may serve as the basis for future research on a better approach to perpetuating histopathological changes beyond 24 h, and the data reported here could also serve as a background for future studies on the mechanism of *P*. *gingivalis* OMV-stimulated neuronal injury in adult zebrafish. It may also provide insight into a better route of administering *P. gingivalis* and its virulence factors.

## 2. Results

### 2.1. Extraction of P. gingivalis OMVs and Authentication

The volume of OMVs recovered after ultracentrifugation was approximately 1.5 mL of OMVs from 2 L of *P. gingivalis* grown to the late log phase:1.5:2000 = 0.075%

Western blotting analysis was completed on the OMV sample (30 & 60 µg loaded) in which the arginine-specific gingipains (RgpA) were detected (western blot result is shown in Appendix A). The arginine-specific gingipain, RgpA, is a complex protein with approximately 95 kDa to 110 kDa. This confirms that the derivation technique was accurate and precise and the neurotoxin used in the study was an OMV.

### 2.2. LD50 of OMVs in Adult Zebrafish

Figure 1 shows that there is a relationship between log dose and probit; i.e., as the log dose increases, the probit increases. The LD50 was determined by plotting the log dose against the probit. From the curve equation, the LD50 is determined by substituting the value of Y at 50% to get the equivalent value of X. From the calculation, the LD50 was found to be 21 µg.

### 2.3. OMV Triggers Increased NO Production in Zebrafish

In Figure 2 below, the graph depicts the NO levels in the zebrafish treated with the OMV and the corresponding control groups in the time points shown in the graph. There was a significant increase in NO levels in the OMV zebrafish group at 45 min compared with the untreated control at the same time point (*p* < 0.05). However, from 6 h till the end of the study, the NO levels were statistically not different from their corresponding control groups.

### 2.4. OMV Does Not Affect Caspase 3 and 9 Activity

Figure 3 shows the activities of apoptotic enzymes, caspase 3 (A) and caspase 9 (B) at 0, 6, 12, 24, 48, 72 h, and 7 days in the OMV-treated zebrafish and the untreated control groups. When the means of the treated and untreated were compared statistically, it was observed that there was no difference between the caspases 3 and 9 groups and their corresponding control groups at all time points. This suggests that the two apoptotic enzymes displayed no activity throughout the study.

### 2.5. P. gingivalis OMV Stimulates Production of IL-1β in Zebrafish

Figure 4 shows the concentration of IL-1β in the Y-axis at seven time points as depicted in the X-axis of the graph. Statistical analysis shows no difference between the OMV-treated zebrafish and the control at 45 min and 6 h time points with *p* values of 0.84 and 0.828, respectively. Nevertheless, there was a significant rise in the concentration of the OMV-treated zebrafish against the untreated control at 12 h (*p* = 0.0384). Post 12 h, there was no statistical difference between the concentration of IL-1β in the brains of the treated zebrafish and the untreated groups.

### 2.6. P. gingivalis OMV Activates Release of IL-6 in Zebrafish

Figure 5 below shows the concentration of IL-6 in OMV-induced zebrafish and the corresponding control group injected only with normal saline. It was observed that the OMVs did not affect the concentration of IL-6 at 0, 6, and 12 h with *p* values of 0.975, 0.308, and 0.425, respectively (*t*-test comparing the treated zebrafish with the control). However, the effect of the OMV on the concentration was significantly higher than the untreated control at 24 h (*p* = 0.0145). Nonetheless, the impact of the OMV on the concentration of IL-6 was not sustained beyond the 24 h time point, as the concentration of IL-6 in the OMV-treated zebrafish and the untreated control group were not statistically different from each other at the 48 h, 72 h, and seven days’ time points.

### 2.7. P. gingivalis OMV Triggers Biosynthesis of PGE2 in Zebrafish

In Figure 6 below, the effect of the OMV on the synthesis of PGE2 was noticeably high at the early stage of the study, with the OMV-induced zebrafish exhibiting significantly higher concentrations of PGE2 than the control group at the 45 min time point. Nevertheless, the other time points showed that concentration of PGE2 in the OMV-induced zebrafish and the control groups were not significantly different.

### 2.8. OMV-Induced Histopathological Changes in the Brains of Zebrafish

Brain histology with hematoxylin and eosin staining, as shown in Figure 7A–D, revealed control (A) and the OMV-treated zebrafish at the 45 min time point (B), the optic tectum with well-organized cells, and there was no congestion of blood vessels. The astrocytes are not more than usual. The typical brain architecture was retained in (A) and (B). However, in the OMV-treated zebrafish brain at the 6 h time points (C), apart from the congestion of the apparent blood vessels (black arrow), other cells and the brain’s architecture remained normal, undistorted, and similar to the control group. Nevertheless, in the OMV-treated zebrafish at the 12 h time point (D), in addition to congestion of blood vessels (black arrow), the astrocytes are more than usual (dark red circle), i.e., astrocytosis is beginning to set in.

Similarly, at the 24 h time point (E) in Figure 7E–H, the architecture of the tectum is beginning to change, the astrocytes and glial cells are becoming active, and there are signs of edema (yellow arrow), spongiosis, and necrosis. Post 24 h, i.e., from 48 h till the end of the study at seven days, the typical brain architecture was observed, the cells were not crowded, and there was no congestion of blood vessels. The brain’s architecture and the cells’ arrangement at the 48 h, 72 h, and 7-day time points were not different from the control groups.

## 3. Discussion

The brain’s innate immune response can result in neuroinflammation, which can manifest in two ways: (1) acutely, when inflammatory mediators are transiently expressed, and (2) chronically, when the resolution of inflammation takes a long time. The latter phase, which gradually contributes to the loss of neuronal cells, is characterized by persistent, low-grade inflammation brought on by the glial cells’ proinflammatory cytokine release. Unresolved systemic inflammatory stimulation results in a condition where hypersensitive, continuously active microglia react aggressively to new immunological triggers [29,30]. Research has shown that the brain experiences a proinflammatory response when LPS activates TLR4 in astrocytes and microglia [22,31].

A single intraperitoneal injection of 18 µg of *P. gingivalis* OMVs triggered an inflammatory response in the brain of adult zebrafish from the 45 min time point up to the 24 h time point. While nitric oxide (NO) and prostaglandin E2 (PGE2) levels were significantly high at the 45 min time points, the histopathological alterations due to the OMV treatment were apparent between the 6 h time point and the 24 h time point. However, IL-1β and IL-6, the two inflammatory cytokines, were expressed at the 12 h and 24 h time points, respectively. The activity of astrocytes and glial cells between the 12 h and 24 h time points, which culminated in astrocytosis, spongiosis, edema, and necrosis, are pointers to the fact that there is a strong relationship between these histopathological changes and the expression of inflammatory cytokines. We have demonstrated here that the OMV-induced neuroinflammation was due to a series of physiological events that are time point dependent in which astrocytes seemed to have played a central role by altering the inflammatory process before the 48 h time point.

NO levels were elevated only in the first 45 min because it is a short-lived gas that influences events that are synonymous with the beginning of the inflammatory response, such as activation of vasodilation, enhancement of leucocyte adhesion, and vascular permeability [32,33]. In the CNS, glial cells generate moderate NO in response to inflammation. Excessive NO synthesis can aggravate neuroinflammation, neuronal death, and uncontrolled tissue damage [32,34]. Interestingly, IL-1β and IL-6 were expressed at different time points. While IL-1β is a proinflammatory cytokine, IL-6 has proinflammatory and anti-inflammatory properties [35]. However, protein expression of IL-6 at a 24 h time point was a control mechanism by the anti-inflammatory phenotype of IL-6 to regulate the inflammatory response and restore homeostasis. Moreover, expressions of IL-1β and IL-6 correspond with the crowding of astrocytes at the two time points. This suggests that the astrocytes must have released the two cytokines in response to the neurotoxic effect of the OMV.

Infectious agents enter host organisms, triggering the immune system to activate a complex defense mechanism that encompasses the activation of an array of immune cells alongside the synthesis of diverse signaling molecules, notably cytokines [36]. Pro-inflammatory cytokines, like IL-1β and IL-6, initiate the immune response, promoting cell recruitment and vascular permeability [37]. Conversely, anti-inflammatory interleukins, like IL-10 and TGF-β, limit inflammation and prevent tissue damage by suppressing pro-inflammatory cytokines [38]. Recent studies have shown that some anti-inflammatory cytokines are linked to controlling specific proinflammatory mediators. For instance, IL-4 blocks the synthesis of cyclooxygenase-2 (COX-2) and inducible nitric oxide synthase (iNOS), which produce PGE2 and NO, respectively. [39], Likewise, they limit the protein expression of pro-inflammatory chemokines such as IL-8, CCL2, CCL3, CCL4, and CCL5. IL-10 has also been reportedly efficient in inhibiting IL-6 and IL-1β [38].

The relationship between histopathological alterations and the protein expression of inflammatory markers shown in this study is supported by the outcomes of some identical studies. For instance, a rodent model that simulated cerebral venous congestion resulting from arteriovenous anastomosis also demonstrated symptoms of cognitive impairment [40,41]. In addition, substantial evidence supports the association between mice experiencing cerebral venous congestion, blood-brain barrier (BBB) disruption, increased extravasation of IgG (and potentially other plasma components), and an exacerbated neuroinflammatory response [42,43]. This is evident through the increased presence of activated microglia and the upregulation of inflammatory mediators within the hippocampus. It has also been observed that pro-inflammatory cytokines, chemokines, proteases, and reactive oxygen species produced by activated microglia contribute to neuronal dysfunction [43].

Several research efforts focused on re-evaluating the importance of inflammation in neurodegeneration are ongoing, and neuroinflammation is recognized as a critical factor in the development of neurodegenerative disease [4]. Despite the neurodegenerative diseases, including Alzheimer’s disease (AD), Parkinson’s disease, multiple sclerosis, and Huntington’s disease, having different pathogenic mechanisms such as genetic differences and varying protein aggregates, the common denominator among them is chronic neuroinflammation. The endogenous pathologic protein aggregation in each of the disorders is capable of triggering neuroinflammation, thereby prompting neurodegeneration [44].

The *P. gingivalis* OMV was adopted for this study because the OMV contains gingipains, LPS, and DNA, all of which have been detected in AD patients [45]. The *P. gingivalis* OMV has a crucial advantage over the *P. gingivalis* cells because of its ability to enter the brain [46]. This attribute is reliant on three critical factors. Firstly, the *P. gingivalis* OMV has a numerical advantage over the *P. gingivalis* cells [45]. Secondly, they are smaller, with a diameter of 80 nm, compared with the *P. gingivalis* diameter of 600 nm [12]. Thirdly, the *P. gingivalis* OMV contains gingipain and LPS, which have been reported to cause AD-like pathological changes [47]. The short-term neurotoxic effect of the OMV in this study does not suggest they were degraded by the fish lysosomal enzymes. OMVs are lipid-membrane-bounded nanoparticles [48], and they have been used successfully to deliver drugs and vaccines [49,50]. OMVs penetrate host epithelial cells through a variety of endocytic pathways, which include both lipid raft-dependent and lipid raft-independent endocytosis, micropinocytosis, clathrin-mediated endocytosis, caveolin-mediated endocytosis, and dynamin-dependent internalization [49].

OMVs derived from *P. gingivalis* have been shown to possess the ability to transmit the required signals for initiating and activating the inflammasome. This condition could guarantee a robust activation of pyroptosis in macrophages, and the increased levels of gingipains identified in the OMVs compared to their parent cells could explain this discovery [51]. Furthermore, the enzymatic activity of gingipains has been recognized as a catalyst for the activation of inflammasomes and the triggering of cell death in different types of cells [52,53]. In contrast to their parent cells, OMVs [54] lack nucleoside-diphosphate kinase (NDK), which means they cannot rely on NDK suppression to prevent ATP/ROS-mediated inflammasome activation [12].

Bacterial OMVs transport lipopolysaccharide (LPS) to the cytosol, activating inflammasomes and pyroptosis in macrophages, as demonstrated by Vanaja et al. in 2016 [55]. Therefore, it is reasonable to suggest that the high concentration of gingipains in OMVs, together with their ability to access the cytosolic inflammasome complex, is responsible for their notable capacity to trigger pyroptotic cell death in macrophages [53]. The activation of pyroptotic cell death by *P. gingivalis* OMVs (as well as other pathogens like *A. actinomycetemcomitans*) could help explain the increased levels of lactate dehydrogenase (LDH) detected in the saliva of individuals with periodontitis [56].

Regarding the apoptotic enzymes (caspase 9 and caspase 3), it was observed that they did not show activity at any of the time points. All of the wild-type strains of *P. gingivalis*, namely A7A1-28, ATCC 49417 (the strain used for this study), and W83, exhibited the ability to hinder the activation of caspase 3 induced by chemical means. The strains that impede apoptosis have unique characteristics. They don’t produce long (FimA) and short (Mfa) fimbria [54,55]. Interestingly, these fimbriae are crucial in causing apoptotic cell death in the host cells by inactivating caspase 3 [49]. Since our strain belongs to the FimA and Mfa non-producing strain, we can infer that our OMVs lacked these two fimbriae. However, a study reported the capacity of *P. gingivalis* to inhibit the activation of caspase 9 [57,58]. Although caspases 3 and 9 are apoptotic, other apoptotic caspases depend on caspase 9 for their activity. In other words, they cannot be activated if caspase 9 is not activated. So, caspase 9 is referred to as an initiator apoptotic enzyme, while caspases 3 and 6 are tagged as executioner enzymes [59]

The cascade of events in this study shows that OMV-induced neuroinflammation in adult zebrafish is a process involving several inflammatory mediators that are time-point dependent. While PGE2 and NO production were significant at a 45 min time point, blood coagulation was the only event observed at 6 h. However, there was an apparent relationship at the 12 h and 24 h time points between the expression of inflammatory cytokines and histopathological alterations such as blood coagulation, astrocytosis, necrosis, edema, and spongiosis.

## 4. Materials and Methods

### 4.1. P. gingivalis Derived OMV Preparation

*P. gingivalis* OMV derivation protocol was modified according to Ho et al. (2015). *P. gingivalis* was grown for 5 to 7 days to reach the late log phase. After that, the culture was subjected to low- and high-speed centrifugation, and the OMV pellet was harvested. The OMV sample was subjected to a western blotting analysis by targeting gingipain R1 (RgpA) [60].

Following the utilization of the Bradford reagent to quantify the protein content in the sample, the total protein (30 µg and 50 µg) was segregated using sodium dodecyl sulfate-polyacrylamide gel electrophoresis (SDS-PAGE) utilizing the subsequent methodology. The separated proteins were transferred to the PVDF membrane by sandwiching the gel and the membrane between the sponge and filter papers in the following order (sponge–filter paper–gel–membrane–filter paper–sponge), starting from the black side of the gel holder cassette to the clear side. The cassette was placed in the electrode assembly, and an electric current was made to run through the cassette. After successfully transferring proteins to the membrane, they were treated with a Rabbit anti-*P. gingivalis* RgpA Polyclonal antibody (Cusabio, Houston, TX, USA) at a concentration of 1:1000, rinsed five times, and then incubated with horseradish peroxidase-conjugated goat anti-rabbit immunoglobulin G secondary antibodies (Gene Tex, Irvene, CA, USA) at a concentration of 1:2000. The samples underwent six cycles of washing in PBST, and the blots were visualized using the Chemidoc™ XRS system (Bio-Rad, Hercules, CA, USA).

### 4.2. Fish Husbandry and Care

All procedures were carried out at the Zebrafish Laboratory, Faculty of Medicine, Universiti Kebangsaan Malaysia, in compliance with ethical approval by the Animal Ethics Committee of Universiti Kebangsaan Malaysia (FPG/2021/NURRUL SHAQINAH/28-JULY/1189-JULY-2021-JUNE-2024). The fish were housed in a 6L 3-dimensional transparent acrylic aquaria tank with a 3-dimensional shape (L, 27 cm × W, 12 cm × H, 15 cm) connected to an automated circulating aquatic system. The fish were stocked at a density of 5 fish per liter and acclimatized for at least 7 days before any experimental procedure was performed. All fish were fasted 24 h before the experimental method was conducted. The fish were maintained at 27 °C ± 0.5 °C and at a 14:10 h light-dark cycle. A twice-daily feeding arrangement (during and pre-experimental) was practiced with a commercial flaked food (New Life Spectrum, Thera + A, Underwood, IN, USA) throughout the experimental duration.

### 4.3. Intraperitoneal Injection of the P. gingivalis-Derived OMV

Fully acclimatized fish were selected randomly and assigned to control and treatment groups. Fish were anesthetized by transferring them with a hand net one at a time to a beaker containing 0.0035% Benzocaine. Immediately the fish slowed their movement of the operculum with no response to touch. The anesthetized fish was moved to an improvised surgical bed that had been slit in the middle. The fish were positioned belly up, followed by carefully inserting the needle between the pelvic fins. The OMV group received 18 µg of OMVs, while the control group received normal saline. The injections were administered using a micro syringe (Hamilton Gastight syringe, 1700 series, Luer tip, Hamilton Company, Reno, NV, USA). After injection, the fish was transferred to a beaker containing one-third of the system water and placed in the designated group accordingly. However, sometimes water needed to be swirled towards the gills to hasten recovery or assist struggling fish to attain full recovery [61,62].

### 4.4. Median Lethal Dose Determination

After a week of acclimatization, including 24 h of fasting, 160 zebrafish were divided equally into eight groups to receive different doses of the OMV (40, 30, 20, 15, 10, 5, 2.5, 1.0 µg). The fish were anesthetized in a beaker containing 0.0035% Benzocaine. Once they showed reduced motor functions, each fish was injected with their respective OMV doses intraperitoneally (using a 30 G syringe with a Hamilton microsyringe) (Hamilton Company, Reno, NV, USA). After the OMV intraperitoneal injection, the zebrafish were put in a beaker with one third of the system water for recovery from anesthesia. Fresh system water is swirled to the fish’s gills until the fish starts swimming freely up and down the beaker [61]. Fish that failed to recover fully from anesthesia were not selected for the experiment. The number of deaths in each group was monitored for seven days. The health of experimental fish was maintained by keeping them in freshly cleaned tanks containing system water every two days or immediately after an incidence of mortality in a tank. The LD50 was determined using the Miller and Tainter methods (a simple regression method in which the probit is plotted against the log dose) [63].

### 4.5. Determination of NO Levels, Caspase 3 and 9 Using Colorimetric Assay

A total of 252 zebrafish were selected for nitric oxide (NO), caspase 3, and caspase 9 measurements. Each OMV group and the control group received 18 µg of OMVs and 10 µL of normal saline via intraperitoneal route, respectively, as described above. Experimental fish were euthanized by immersing them in ice water at designated time points (45 min, 6, 12, 24, 48, 72 and 168 h), which was followed by brain dissection and homogenization according to the kit manufacturer’s instructions (Elabscience, Houston, TX, USA).

### 4.6. Determination of Concentrations of Interleukin 6 (IL-6), Interleukin 1 Beta (IL-1β), and Prostaglandin E2 (PGE2) Using ELISA

In experimental procedures using 126 adult zebrafish, concentrations of IL-6, IL-1β, and PGE2 were estimated from the homogenate of brain tissues of zebrafish induced with the OMV and the uninduced control group at all seven time points (45 min, 6, 12, 24, 48, 72, and 168 h) using the ELISA protocol according to the manufacturer’s instructions (ELK Biotechnology, Denver, CO, USA).

### 4.7. Evaluation of Zebrafish Brain Histology

After the expiration of each time point, the whole of the zebrafish head was dissected for brain histological examination. (The head was fixed in a 10% formalin solution for 48 h. The tissue samples were processed and stained using routine H&E protocol. The histology images were analyzed using a light microscope (Olympus, Center Valley, PA, USA).

### 4.8. Statistical Analysis

The LD50 determination was evaluated using the Miller and Tainter method. The other results were presented as each experimental group’s mean ± standard deviation (SD), determined using an unpaired *t*-test. The results were selected from at least three independent experiments carried out in triplicate, and *p*-values of 0.05 or less were statistically significant.

## Figures and Tables

**Figure 1 ijms-25-11025-f001:**
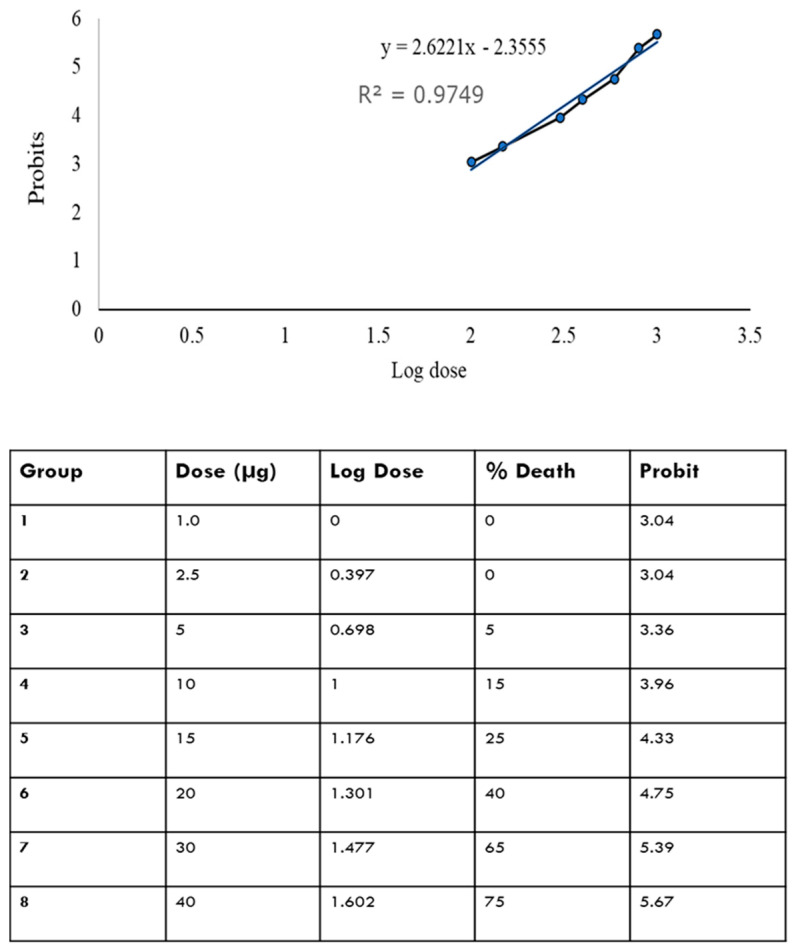
Log10 dose versus the probit. The probit value is the standardized conversion value derived from the corrected mortality percentage as adapted from Finney’s table.

**Figure 2 ijms-25-11025-f002:**
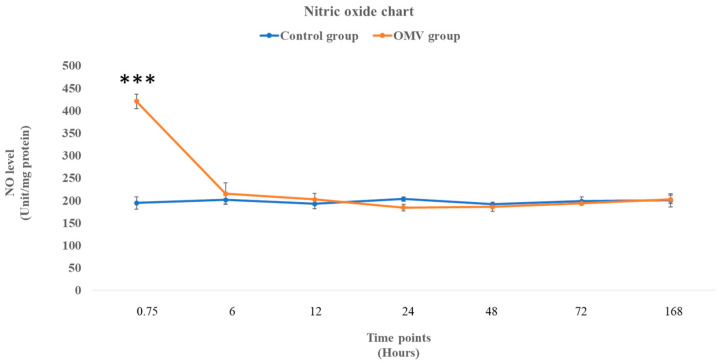
The level of NO in the brains of OMV-treated and the control group of zebrafish. Values are mean ± standard error of the mean (SEM) (*n* = 6/group). *** This symbol represents that the treated zebrafish group was significantly different from the control group at *p* < 0.001.

**Figure 3 ijms-25-11025-f003:**
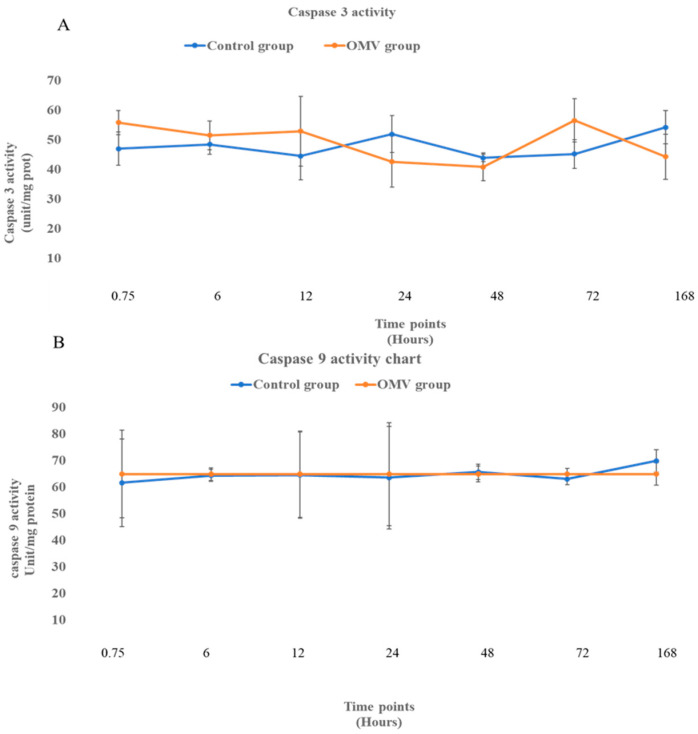
(**A**). The activity of caspase 3 in the brains of the OMV-treated group and the control group of zebrafish. (**B**). The activity of caspase 9 in the brains of OMV-treated and the control group of zebrafish. Values are mean ± SEM (*n* = 6/group). OMV-treated zebrafish were not significantly different from the control group at *p* < 0.05.

**Figure 4 ijms-25-11025-f004:**
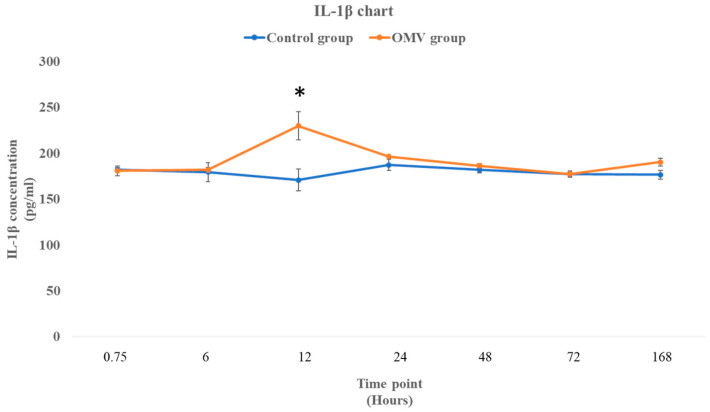
The concentration of IL-1β in the brains of OMV-treated zebrafish and the control group of zebrafish. Values are mean ± SEM (*n* = 3/group). * This symbol represents OMV-treated zebrafish that were significantly different to the control group at *p* < 0.05.

**Figure 5 ijms-25-11025-f005:**
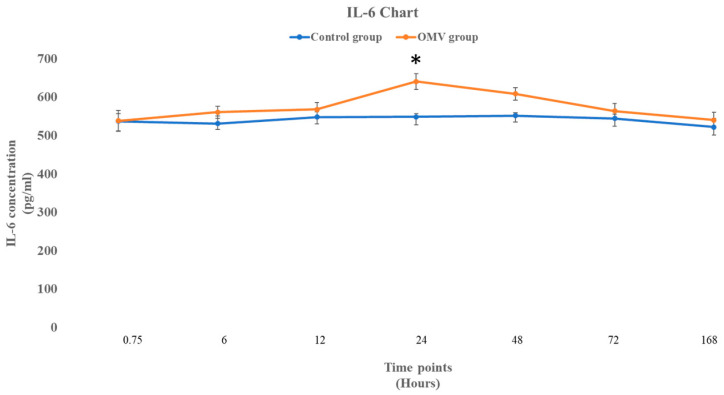
The concentration of IL-6 in the brains of OMV-treated and the control group of zebrafish. The values represent the mean ± SEM (*n* = 3 per group). * This symbol represents OMV-treated zebrafish that were significantly different to the control group, with a *p*-value less than 0.05.

**Figure 6 ijms-25-11025-f006:**
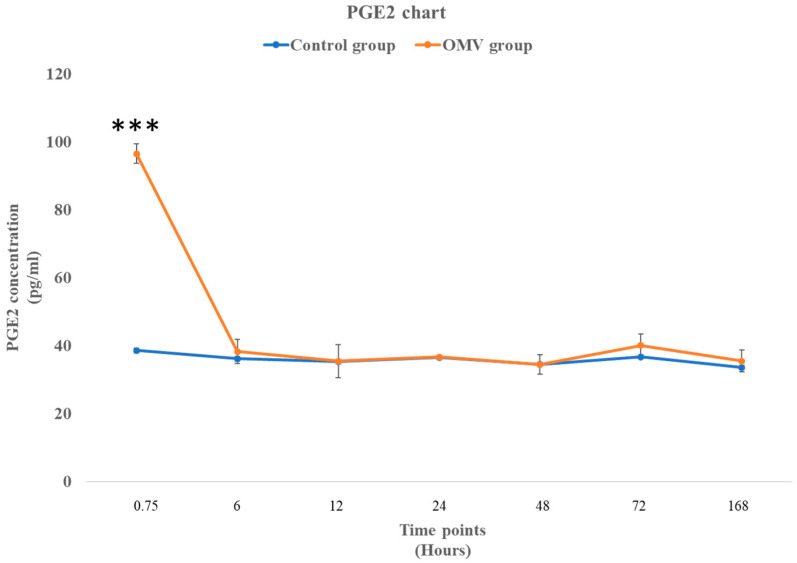
The concentration of PGE2 in the brains of the OMV-treated and the control group of zebrafish. The values represent the mean ± SEM (*n* = 3 per group). *** The zebrafish group treated with OMVs was significantly different to the control group, with a *p*-value of less than 0.001.

**Figure 7 ijms-25-11025-f007:**
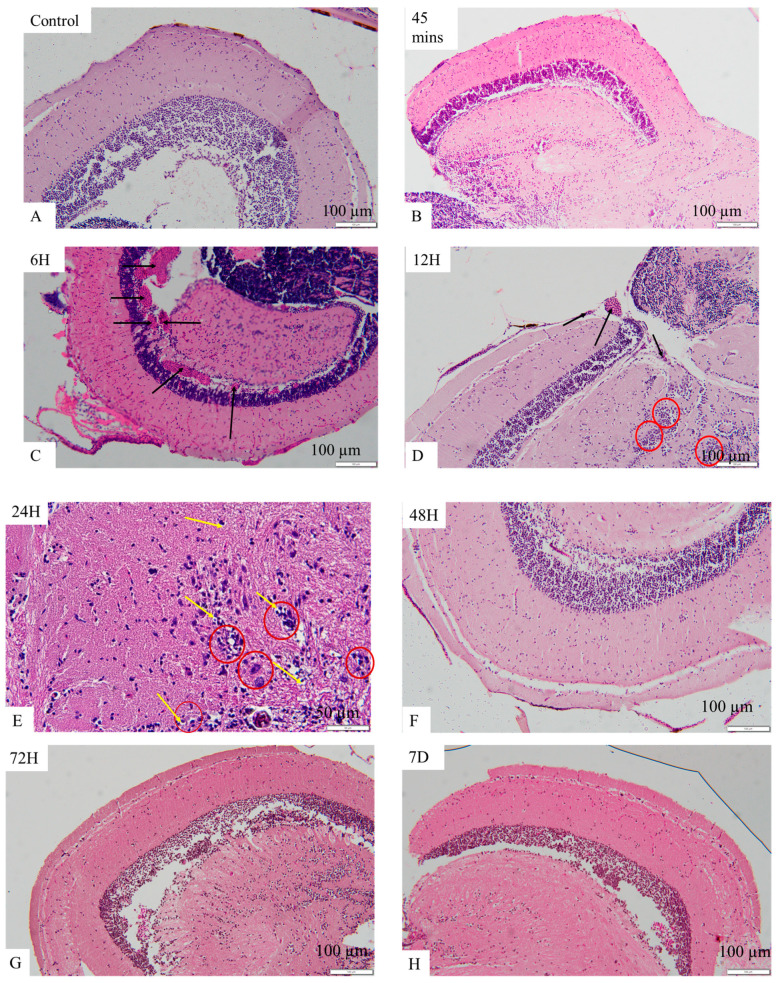
Histological changes in the brain of zebrafish. (**A**) the control group that is injected with normal saline only; (**B**) the brain of zebrafish injected with the OMV at the 45 min time point; (**C**) the brain of zebrafish injected with the OMV at the 6 h time point; (**D**) the brain of zebrafish injected with the OMV at the 12 h time point; (**E**) the brain of zebrafish injected with the OMV at the 24 h time point; (**F**) the brain of zebrafish injected with the OMV at the 48 h time point; (**G**) the brain of zebrafish injected with the OMV at the 72 h time point; (**H**) the brain of zebrafish injected with the OMV at the 7 days’ time point.

## Data Availability

Data are contained within the article.

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
