# Peer review of "Intraperitoneal Injection of the Porphyromonas gingivalis Outer Membrane Vesicle (OMV) Stimulated Expressions of Neuroinflammatory Markers and Histopathological Changes in the Brains of Adult Zebrafish"

_ijms, 2024, doi:10.3390/ijms252011025_

Round 1
Reviewer 1 Report
Comments and Suggestions for Authors
Comments for the authors:
A single dose intraperitoneal injection of P. gingivalis outer membrane vesicle (OMV) stimulated expressions of neuroinflammatory markers and histopathological changes in the brain of adult zebrafish” is a very interesting work, and I would like to congratulate the authors for it. Determining the effect of P. gingivalis OMV on the production and expression of inflammatory markers in adult zebrafish may help in understanding histopathological changes in the central nervous system.
I would like to make just a few remarks:
- In the captions of figures 7a and 7b, please add the scale bar dimensions, as they are not readable from the images.
- Also, in these figures, I would suggest increasing the thickness of the arrowhead.
- It would be better to elaborate more on the introduction by providing more studies present in the literature between P. gingivalis and periodontitis, which is the research's purpose.
- I recommend adding the role of the markers used at least in the results, if not a brief explanation in the introduction. For example: “caspase 3 is an apoptotic marker, its activation indicates... which is why we investigated...” to present the results with more argumentation.
- The discussion is well argued.
Author Response
- In the captions of figures 7a and 7b, please add the scale bar dimensions, as they are not readable from the images.
-
The scale bar has been made bold
-
Also, in these figures, I would suggest increasing the thickness of the arrowhead.
We have increased the thickness of the arrowhead
It would be better to elaborate more on the introduction by providing more studies present in the literature between P. gingivalis and periodontitis, which is the research's purpose.
The introduction has been rewritten as requested. Line 44 - 66
I recommend adding the role of the markers used at least in the results, if not a brief explanation in the introduction. For example: “caspase 3 is an apoptotic marker, its activation indicates... which is why we investigated...” to present the results with more argumentation.
The inflammatory mediators are mentioned in the introduction as recommended (line 49 – 79)
Reviewer 2 Report
Comments and Suggestions for Authors
This study examined the real-time effects of Porphyromonas gingivalis' outer membrane vesicles (OMV) on neuroinflammation and CNS pathology in zebrafish. Injecting OMV led to rapid increases in NO, PGE2, and later IL-1β & IL-6, accompanied by histopathological changes. However, no apoptotic markers were detected. The results indicate that OMV-induced neuroinflammation and pathology are time-dependent. Nonetheless, numerous revisions are still required to address various issues prior to the article's acceptance.
Major comments
1. The English sentences throughout the entire manuscript are challenging to comprehend due to the low quality of writing. Kindly revise them meticulously.
2. Why does the expression of NO peak at 45 minutes in the OMV group, yet remain largely unchanged after 6 hours? Could you elaborate on the mechanisms that govern this process? Additionally, as time passes, does OMV undergo degradation and metabolism within the zebrafish body? Please explain.
3. Why do IL-1β and IL-6 exhibit different timings of high expression following OMV treatment in zebrafish? Additionally, given that TNF-α is a significant proinflammatory factor, why was it not included in the testing? Furthermore, could you clarify the expression patterns of the anti-inflammatory factors IL-4, IL-10, and TGF-β across various time points? We kindly request a detailed explanation.
4. Please incorporate histological scoring data into Figures 7a and 7b, and also furnish statistics regarding the quantity of astrocytes present at different time intervals.
5. In Figures 7a and 7b, the author observed a significant increase in the number of astrocytes in the OMV group at both 12 and 24 hours, accompanied by elevated expression levels of IL-1β and IL-6. Does this indicate that these two inflammatory factors are predominantly expressed by astrocytes? Please explain.
6. In the Introduction section, kindly present a hypothesis that frames the entire paper.
Minor comments:
1. There are redundant spaces in Line 26 "and histology", please adjust it.
2. Lines 68-71 contain sentences that are excessively long. Kindly revise them to enhance readability. Additionally, please review all sentences throughout the entire manuscript.
3. Line 104, there is a dot "." at the beginning of the sentence, please check.
4. The units of the vertical coordinates of the histograms in Figure 3 A and B should be consistent. Please adjust them.
5. Line 146, a space needs to be added in "ofPGE2", please adjust.
6. In the results section, please specify whether the expression of NO, Caspase3, Caspase9, IL-1β, IL-6, and PGE2 refers to protein or mRNA levels. For instance, you can clarify by writing 'the protein levels of...'.
7. Line 154, please use the abbreviation for haematoxylin and eosin. Additionally, please review the entire manuscript to assess if other terms should also be abbreviated.
8. Lines 260-261, there is a missing period '.' at the end of the sentence. Please review the entire manuscript for similar issues.
Comments on the Quality of English LanguageThe English sentences throughout the entire manuscript are challenging to comprehend due to the low quality of writing. Please revise.
Author Response
The English sentences throughout the entire manuscript are challenging to comprehend due to the low quality of writing. Kindly revise them meticulously
The entire manuscript has been revised.
- Why does the expression of NO peak at 45 minutes in the OMV group, yet remain largely unchanged after 6 hours? Could you elaborate on the mechanisms that govern this process? Additionally, as time passes, does OMV undergo degradation and metabolism within the zebrafish body? Please explain.
This comment has been responded to. (line 239 – 224)
- Why do IL-1β and IL-6 exhibit different timings of high expression following OMV treatment in zebrafish? Additionally, given that TNF-α is a significant proinflammatory factor, why was it not included in the testing? Furthermore, could you clarify the expression patterns of the anti-inflammatory factors IL-4, IL-10, and TGF-β across various time points? We kindly request a detailed explanation.
Response to this request can be found between lines 244 and 263. Regarding TNF-α, the kit we got was beyond our budget. Zebrafish ELISA kit brands are very few.
- Please incorporate histological scoring data into Figures 7a and 7b, and also furnish statistics regarding the quantity of astrocytes present at different time intervals.
Due to insufficient data, we could only perform qualitative analysis by describing the histopathological changes observed. The suggestion is highly appreciated and will be included in future studies.
- In Figures 7a and 7b, the author observed a significant increase in the number of astrocytes in the OMV group at both 12 and 24 hours, accompanied by elevated expression levels of IL-1β and IL-6. Does this indicate that these two inflammatory factors are predominantly expressed by astrocytes? Please explain
Response to this comment can be found between lines 248 and 251 .
Minor comments
- There are redundant spaces in Line 26 "and histology", please adjust it.
It has been adjusted
- Lines 68-71 contain sentences that are excessively long. Kindly revise them to enhance readability. Additionally, please review all sentences throughout the entire manuscript.
It has been revised
- Line 104, there is a dot "." at the beginning of the sentence, please check.
The dot has been removed
- The units of the vertical coordinates of the histograms in Figure 3 A and B should be consistent. Please adjust them.
The units are now the same
- Line 146, a space needs to be added in "ofPGE2", please adjust
Space has been added.
- In the Introduction section, kindly present a hypothesis that frames the entire paper.
The response to this comment can be found between lines 84 and 86
Round 2
Reviewer 2 Report
Comments and Suggestions for Authors
I am generally satisfied with the author's revisions to the scientific research article. He/She has made reasonable adjustments to the article's structure, thereby enhancing its persuasiveness and credibility. Furthermore, the author has provided detailed responses to the comments, showcasing a commendable academic attitude and a rigorous research spirit. Overall, these revisions have significantly elevated the overall quality and academic value of the research article.